# Characterizing drug mentions in COVID-19 Twitter Chatter

**Ramya Tekumalla**     **Juan M Banda**
Department of Computer Science
Georgia State University
{rtekumalla1, jbanda}@gsu.edu

## Abstract

Since the classification of COVID-19 as a global pandemic, there have been many attempts to treat and contain the virus. Although there is no specific antiviral treatment recommended for COVID-19, there are several drugs that can potentially help with symptoms. In this work, we mined a large twitter dataset of 424 million tweets of COVID-19 chatter to identify discourse around drug mentions. While seemingly a straightforward task, due to the informal nature of language use in Twitter, we demonstrate the need of machine learning alongside traditional automated methods to aid in this task. By applying these complementary methods, we are able to recover almost 15% additional data, making misspelling handling a needed task as a pre-processing step when dealing with social media data.

## 1 Introduction

The World Health Organization (WHO) defines the Coronavirus disease (COVID-19) as an infectious disease caused by a newly discovered coronavirus and declared it a pandemic on March 11, 2020 (World Health Organization). As of June 26, 2020, **9,764,997** cases were confirmed worldwide with **492,807** deaths and **4,917,328** cases recovered. Social media platforms like Twitter and Reddit contain an abundance of text data that can be utilized for research. Over the last decade, Twitter has proven to be a valuable resource during disasters for many-to-many crisis communication (Zou et al., 2018; Earle, 2010; Alam et al., 2018). Recently, several works (Lu, 2020; Sanders et al., 2020; Gao et al., 2020) have provided insights on the treatment options and drug usages for COVID-19.

We utilized the largest available Covid-19 dataset (Banda et al., 2020) curated using a Social Media Mining Toolkit (SMMT) (Tekumalla and Banda, 2020b). Version 15 of the Covid-19 dataset was utilized for our experiments since it was the latest released version at the time of experiments. This dataset consists of tweets related to COVID-19 from January 1, 2020 to June 20, 2020. We automatically tagged ~424 million tweets using a drug dictionary compiled from RxNorm (National Library of Medicine, 2008) with 19,643 terms and validated in (Tekumalla et al., 2020) and (Tekumalla and Banda, 2020a).

## 2 Methods

In order to identify drug-specific tweets related to COVID-19, we applied the drug dictionary on the clean version of the dataset. With the presence of retweets, any signal found would be greatly amplified and hence only unique tweets must be utilized to identify the signals. The cleaned version of the dataset consists of **104,512,658** unique tweets with no retweets. We are only using the English language tweets, which are 67% (~70 million tweets) of them. The spacy annotator utility from the Social Media Mining Toolkit (Tekumalla and Banda, 2020b) was utilized to tag the clean tweets. A total of **723,129** tweets were identified as containing one or more drug terms. However, complex medical terms, such as medication and disease names, are often misspelled by users on Twitter (Karimi et al., 2015). Using only the keywords with correct spellings leads to a loss of potentially important data, particularly for terms with difficult spellings. Some misspellings also occur more frequently than others, implying that

they are more important for data collection and concept detection. To identify the misspelled drug tweets, we initially identified the top 200 drug terms from the 723,129 drug tweets tagged. Additionally, early in the stages of this work, a preliminary report from a large study of 1,063 patients showed that hospitalized patients who were administered Remdesivir recovered faster than those who got a placebo (Beigel et al., 2020). However, the version of RxNorm we have utilized for the original drug dictionary did not have Remdesivir since it was an investigational drug and was later added to RxNorm in April 2020 (Bulletin, 2020). We manually added this drug term for our analysis.

In this work, we demonstrate the tradeoff between generating misspellings, using generated word-embedding misspellings, and auto-correcting the spellings at the text level. For this purpose, we employed four different methodologies to acquire additional data. The first methodology involves a machine learning approach called QMisSpell (Sarker and Gonzalez-Hernandez, 2018). The RedMed (Lavertu and Altman, 2019) model's word embeddings were utilized in QMisSpell to generate misspellings. This RedMed model has an embedding size of 64 dimensions with a window size of 7 and created utilizing the health-oriented subset of Reddit. QMisSpell relies on a dense vector model learned from large, unlabeled text, which identifies semantically close terms to the original keyword, followed by the filtering of terms that are lexically dissimilar beyond a given threshold. The process is recursive and converges when no new terms similar (lexically and semantically) to the original keyword are found. A total of 2,056 unique misspelled terms were generated using QMisSpell. We examined all the misspellings and eliminated several terms (example: heroes, heroine generated from the original keyword heroin) which are common terms in the English vocabulary and are not misspellings of drug terms. Post elimination, a total of **1,932** terms are identified as the misspelt terms.

The second methodology utilizes keyboard layout distance for generating the misspellings. For each term, each letter is replaced with the closest letter on the QWERTY keyboard. This process is recursive and ceases when it has looped through every letter in the term. For example, the term cocaine has 68 misspelled terms which vary from xocaine to cocaint. After eliminating the common

vocabulary and duplicates, a total of 15,754 terms are identified as misspelled terms for this methodology. We text tagged the clean dataset utilizing the misspelled terms from the methodologies.

The third methodology employs a spelling correction module called Symspell (Garbe, 2019) which corrects the spelling errors at text level before text tagging. This method is built using the Symmetric Delete spelling correction algorithm which reduces the complexity of edit candidate generation and dictionary lookup for a given Damerau-Levenshtein distance (Damerau, 1964; Levenshtein, 1966). This method generates terms with an edit distance (deletes only) from each dictionary term and adds them together with the original term to the dictionary and searches the dictionary. This has to be done only once during a pre-calculation step. For a word of length n, an alphabet size of a, an edit distance of 1, there will be just n deletions, for a total of n-terms at search time. The Symmetric Delete spelling correction algorithm (Symspell) reduces the complexity of edit candidate generation and dictionary lookup by using deletes only instead of deletes + transposes + replaces + inserts(Norvig, 2007). This methodology requires a frequency dictionary of the common terms. Since we are extracting only English drug related tweets, we used the default English frequency dictionary provided by Symspell and added a drug frequency dictionary generated using ~60 million clean tweets. Additionally, we manually included the top COVID-19 keywords (Shah, 2020) with frequencies in our initial tagging, since these terms are found in large numbers. The Symspell algorithm corrects each term in a tweet text to the closest matching term in the dictionary. The corrected tweet is then sent to the SMMT spacy utility for tagging. A total of 17,686 misspelled drug terms were generated for the top 200 drug terms using the Qmispell and Keyboard layout methods.

The fourth methodology utilizes TextBlob (TextBlob), a Python library for processing textual data. It provides a consistent API for diving into common natural language processing (NLP) tasks such as part-of-speech tagging, noun phrase extraction, spelling correction and more. The spelling correction feature of the TextBlob employs Peter Norvig's spelling correction algorithm (Norvig, 2007) which generates a

candidate model by a simple deletion to a word (remove one letter), a transposition (swap two adjacent letters), a replacement (change one letter to another) or an insertion (add a letter). This method was applied on each tweet text before tagging. Unfortunately, this implementation is highly inefficient, taking around 60 hours of execution time for every 600,000 tweets. We were only able to run this on 20 days of data, and only present results in the computational evaluation table.

All experiments in this work are presented in the following section and focus on the first three methods outlined.

## 3    Results

Table 1 summarizes the text tagging results between the three experimental setups we tested. The overlap of the generated misspellings between QMisSpell and the Keyboard Layout generator is 4.9%, meaning only 33 terms were common between the two misspelling dictionaries. The method Spacy + RxNorm Dictionary tagged a total of 1,483,691 terms. The method Spacy ON Sympell Corrected text + RxNorm dictionary was able to tag an additional 149,260 terms (delta) which the previous method could not tag. These are mostly due to the newly spell-corrected tweets text, note that this difference is not indicative of the whole dataset as it was not all spell-corrected.

| Method | Total number of terms in dictionary | Total tagged term in dataset |
|---|---|---|
| Spacy + RxNorm Dictionary | 19,643 | **1,483,691** |
| Spacy + QMisSpell dictionary (only) | 1,932 | **192,619** |
| Spacy + Keyboard layout generated misspellings dictionary (only) | 15,754 | **135,981** |
| Spacy ON Symspell Corrected text + RxNorm Dictionary | 19,643 | **1,632,951 **** |

Table 1:  Text Tagging results ** note that we only corrected text on tweets for 20 days of the dataset, not the complete set.

Unsurprisingly, Hydroxychloroquine was the most tweeted drug found on the dataset with chloroquine coming in as close second, as shown in Table 2. A total of 1,483,691 terms were tagged from 723,129 tweets. Though there are 19,643 unique drug terms in the dictionary, we only show the top 10 most frequent drug terms tagged on Table 2. Interestingly enough, hydroxychloroquine was not the most misspelled word, chloroquine was, as we can see in Table 3.

| Drug Name | Frequency | Percentage |
|---|---|---|
| hydroxychloroquine | 161,385 | 10.88% |
| chloroquine | 106,377 | 7.17% |
| remdesivir | 52,152 | 3.52% |
| agar | 34,505 | 2.33% |
| oxygen | 28,906 | 1.95% |
| zinc | 22,632 | 1.53% |
| azithromycin | 21,183 | 1.43% |
| vitamin d | 19,067 | 1.29% |
| ibuprofen | 14,765 | 1.00% |
| dettol | 11,660 | 0.79% |

Table 2:  Top 10 Most frequent drugs found using the drug dictionary without including misspellings.

In our exploration of misspellings, we had 23 potential different spellings for Hydroxychloroquine. We have 78 possible misspellings of chloroquine as well. While not all possible misspellings are found in the actual Twitter data, a good combination of them (41%) are actually present, indicating how tricky it is to work with Twitter text data. Terms such as Agar, and Coconut Oil acquired high counts as listed on Table 3. This is due to a few irrelevant terms prevailing in the dictionary. The drug dictionary curated from RxNorm consists of terms from different semantic types (Ingredients, Semantic Clinical Drug Component, Semantic Branded Drug Component and Semantic Branded Drug). The Ingredients semantic type consists of different terms that are technically not drugs but are utilized in creating drugs (Eg: Gelatin, Agar, Coconut Oil).

Such terms were still included in the dictionary due to the granularity they bring to the research, particularly zinc, for which a study proved the addition of zinc sulfate to Hydroxychloroquine and Azithromycin increased the frequency of discharge rates of Covid-19 affected patients and hence may play a role in therapeutic management for COVID-19 (Carlucci et al., 2020). In order to evaluate the Symspell method, we counted the additional tagged terms occurrences from the Spacy + RxNorm experiment, which gave us the number of 'misspellings' fixed that were now identified using the same dictionary. In other words, table 3 shows the additional number of drug terms found when using the misspelling dictionaries (keyboard layout, and QMisSpell methods) and identified during the tagging of the spelling-corrected tweets. Out of the top 200 drug terms we generated misspellings for, over 150 drug term misspelling variants were tagged, with Table 3 showing only the most frequent ones. At a glance, it is surprising that after spell-correcting the tweets, only ~200K additional terms are tagged, but this is intuitive as only drug-related terms are being tagged. If using a general purpose dictionary, we would expect this number to be considerably larger due to the fact that Twitter data is quite noisy and constantly misspelled. The total added term counts can be found in Figure 1. As we can see in both Figure 1 and Table 1, by considering the possibility of misspellings, we find around 15% more data. When dealing with limited text availability, particularly drug terms in Twitter, it is vital to use these types of methods to recover as much signal as we can. For this figure we removed the non-COVID-19 related terms manually and only left the more prevalent and relevant (in literature and news) drug terms to simplify our representation and discussion.

| Keyboard Layout Method | | QMisSpell Method | | Symspell Method | |
|---|---|---|---|---|---|
| **Drug Name** | **Freq.** | **Drug Name** | **Freq.** | **Drug Name** | **Freq.** |
| cloroquine | 34,522 | hydroxychloroquine | 13,500 | cloroquine | 14,175 |
| vitamin a | 21,285 | meted | 10,424 | hydroxychloroquine | 11,580 |
| remdesivir | 16,981 | allegra | 10,339 | azithromycin | 10,528 |
| agar | 13,935 | ibuprofen | 8,640 | nicotine | 7,054 |
| hydroxychloroquine | 6,216 | propane | 6,984 | doral | 6,566 |
| doral | 3,496 | coconut oil | 4,812 | cloroquine | 4,620 |
| nicotine | 3,081 | agar | 2,518 | aleve | 2,004 |
| cocaine | 1,890 | azithromycin | 1,965 | septa | 1,914 |
| zinc | 1,424 | aluminum | 1,633 | remdesivir | 1,912 |
| septa | 1,006 | acetaminophen | 1,263 | vitamin a | 1,437 |

Table 3: Most frequent drugs found using the misspelled dictionaries

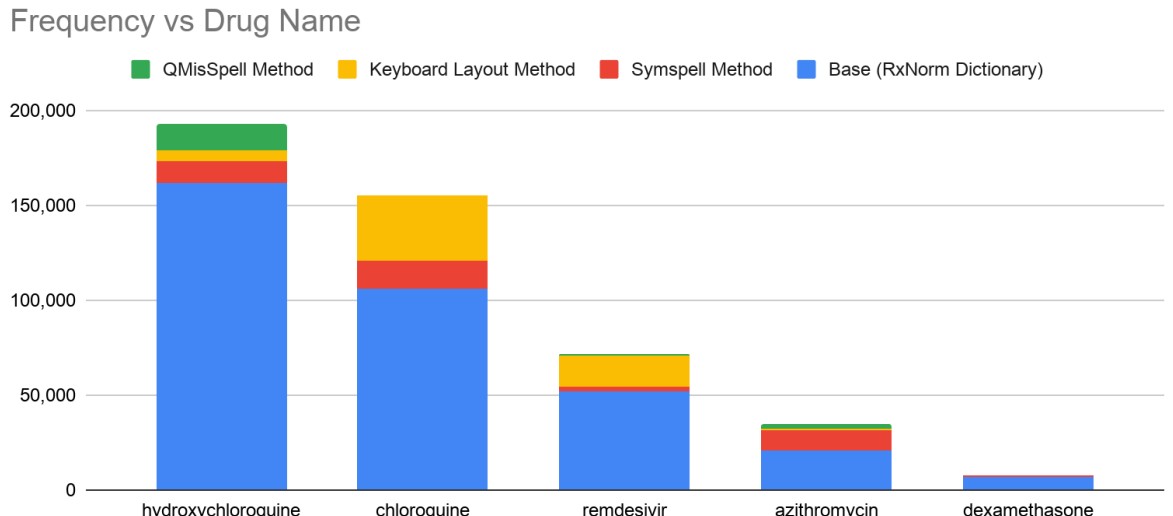

Figure 1: Drug Term occurrences in our dataset

In order to evaluate the cost-benefit of using the misspelling methods to get additional data, we timed the execution of the components of the methods, from generation of dictionaries (where applicable) to total and average time taken for text tagging.

| | Generation time (ms) | Total Tagging time (min) | Avg Tagging Time (per 600,000 tweets) |
|---|---|---|---|
| **Base (RxNorm Dictionary)** | 100 | 6,930 | 45 |
| **Keyboard Layout Method** | 2 | 6,314 | 41 |
| **QMisSpell Method** | 200 | 924 | 6 |
| **Symspell Method** | NA | 166,320 | 1,080 |
| **TextBlob** | NA | 72,000 | 3,600 |

Table 4: Misspelling method execution time analysis

The first item to note is that for both the keyboard and QMisSpell methods, these times need to be added to the baseline times as they are executed independently in addition to the base text tagging. However, they are not even close in terms of computational expense to the Symspell or TextBlob spell correction, which are quite expensive. For TextBlob, we only processed 20 days of data before aborting as this package took

on average 60 hours of execution time per 600,000 tweets.

In order to show the overlaps of terms tagged with each of the different methods explored, Figure 2 shows the total overlap in percent between the new terms tagged. There is a 35% overlap between the 3 methods since there are many generated terms that are produced by all methods, these are usually the more frequent ones or typical ones. Showing that a brute force approach (keyboard based method) might not be ideal to generate misspellings on large sets of terms as it would add considerable computational expense.

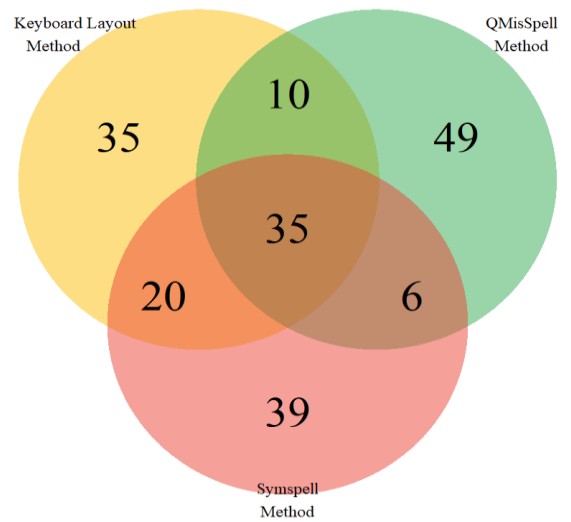

Figure 2: Tagged misspelled terms overlaps between different methods. Note that this is in % points.

Finally, in Table 5 we quantify the individual and combined gain from each of the methods when it comes to recovering additional data points. Note that this table includes all 200 most frequent terms we generated misspellings for, not just the most common COVID-19 drug terms. The 14.99% constitutes the additional number of terms that can be identified when using the outlined techniques for misspelling identification, in addition to the original 1,483,691 terms identified.

| | Additional Terms Identified | Percentage Increase |
|---|---|---|
| Keyboard Layout Method | 132,083 | 8.90% |
| QMisSpell Method | 75,788 | 5.11% |
| Symspell Method | 89,592 | 6.04% |
| Total | 222,418 | 14.99% |

Table 5: Additional terms identified

## 4 Conclusions

In this work, we focused on extracting discourse related to the potential drug treatments available for COVID-19 patients. While seemingly a trivial task, we show that enough consideration is needed for the proper way to deal with the constant misspellings found in Twitter data. We have shown that with a combination of methods we can identify around 15% additional terms, which would have been lost. With data being quite limited and not easily available, it is important to apply the proper techniques to identify the largest subset of data possible. We provide a quantifiable evaluation by generating misspellings utilizing two different methods and automated spell check using Symspell. While the keyboard layout method is a fully automated method to generate misspellings, the QMisSpell method generates misspellings based on language models. We have shown that this work is indeed needed to be performed in order to make a thorough evaluation of the discourse regarding potential drug treatments for COVID-19 patients.

## 5 Future Work

Out of the scope of this work, we theorize that using sentiment analysis and stance detection, we can also identify how users respond to these drugs and identify the drugs that help with some symptoms. With careful analysis, this data can be utilized to monitor the perception of theorized treatment options.

Unfortunately, not many pre-trained embedding models related to drug research are readily available. The language model used in this research was able to generate misspellings for 80% of the top 200 terms. It is immensely difficult to obtain drug related data to create a language model to generate misspellings for all terms due to unavailability of the data. In the future, we would like to explore the usage of deep learning models like Bio-BERT or BERT for the error correction/misspelling generation.

## Acknowledgments

Part of this research was developed during the COVID-19 Biohackathon April 5-11, 2020. We thank the organizers for coordinating the virtual hackathon during the COVID-19 crisis. We would like to thank Stephen Fleischman and HP labs for providing us with server access to perform our experiments during our research server downtime.

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
