# OpenReview forum: "Characterizing drug mentions in COVID-19 Twitter Chatter"
_EMNLP/2020/Workshop/NLP-COVID — NLP-COVID19-EMNLP Poster_

### Official Review · AnonReviewer1 · 2020-09-08
**Clear short paper on Twitter discourse mining for drug mentions discovery**

**Rating:** 7
**Confidence:** 3

**Review:**

In this paper the authors mine the COVID-19 chatter tweets previously released to identify the usage of COVID-19 drugs related vocabulary. While the paper is interesting, well-written and concise, the authors over-sell some of its aspects.

The paper is straightforward, yet interesting and I would say it would be easily understood by a wide audience. The number of experiments are adequate, the results are clearly presented and the discussion of each of the steps of the methodology makes it easy to follow.

On the downside, starting by the title, I considering it misleading as the paper solely addresses the use of drug-related terms in tweets. The sole  use of COVID-19 drugs-related vocabulary is not an indicator of a possible treatment being followed by the author of a given tweet. The same idea is further reinforced in the conclusion, which refers to potential drug treatments available for COVID-19 patients. However, the experiments led by the authors only take into consideration drug vocabulary frequencies, so I fail to see how are these indicative for potential drug treatments. The work presented is of interest and clearly exposed and should be presented as it is: a COVID-19 drug terms frequency on Twitter chatter. The abstract and introduction also lead the reader into thinking a 424 million tweets corpus was mined and annotated, however the corpus size is considerably smaller (the clean corpus has approx. 100 million tweets, out of which only 67% are in English and were considered for this study). Other remarks: in Table 1, what is delta in the last row? Could the authors elaborate on the differences of the drugs found using with the 3 different spelling methods? I would expect the list of drugs to be more similar, but with different frequencies.

---

> ### Author Response · Authors · 2020-09-26
> **Concise and very on-point feedback**
>
> We would like to thank the reviewer for the through and well described review. Here are our comments on your main concerns:
>
> Indeed, the title is unfortunately misleading and we will certainly change it to better represent the paper. Our idea with regards potential drug treatments was to find the drugs/terms that are appearing with high frequencies but are not in the radar of clinicians, thus potentially finding something that people are talking about/taking that might be potentially important. Unfortunately, we didn't find many interesting things with higher frequencies however, something might be buried in the very long tail of less frequently identified drug terms. With regards to the dataset, we do in fact have all the dataset tagged with drug terms, but decided on filtering this to focus more on the non-retweets and English only drug chatter. We will clarify this on the text. Thanks for pointing this out.
>
> Other remarks:
> 1) The delta represents the additional terms tagged when using spell correction on a limited set of tweets.
> 2) We will add this discussion on the difference between methods, as other reviewers have pointed out that is needed.
>
> Thank you again for the detailed feedback.

---

### Official Review · AnonReviewer2 · 2020-09-18
**Interesting work but requires further analysis and validation**

**Rating:** 5
**Confidence:** 4

**Review:**

The paper points out that spelling correction is an important step in preprocessing covid-19-related tweets and enables identification of additional data. The authors explore several approaches to correcting spelling mistakes in drug names using a corpus of covid-19 related tweets. The results suggest that several spelling correction techniques should be applied together to comprehensively address the problem.
At the same time, my feeling is that the title of the paper does not correspond to its content. The paper does not provide a characterisation of potential treatments but rather lists the most commonly mentioned drugs. There is no further characterisation of the context in which those drugs are mentioned. Moreover, both the title and the abstract put emphasis on using machine learning, however, only 1 method out 4 is machine learning-based.
Overall, the format of this study seems more appropriate for a short paper. For a long paper, I would expect the authors to further analyse why the implemented algorithms give different results, what potential errors might be introduced as well as show some validation of the approach.

Areas for improvement:

- When describing the second method used to generate misspellings the authors state “For each term, each alphabet is replaced with the closest alphabet on the QWERTY keyboard.” It is not clear what the authors mean by an “alphabet”, I am assuming it means a “letter”? How the closest letter on the keyboard is defined? Do variations include digits? Where is the recursion in this approach? I suggest the authors either cite a related research paper or provide a more detailed explanation.
- Table 2 shows the number of mentions of the top 10 drugs. It is unclear whether these numbers were calculated after applying spelling correction techniques or not.
- I suggest moving the clarification for the data reported in Table 3 from the bottom of p.3 to when the table is first mentioned.
- It would be great to see more analysis of why the overlap between the three methods is only 15%, ideally, with specific examples.
- Table 5 provides the numbers of additional terms identified using the described methods. How many additional tweets does it allow to recover? How can the authors ensure the additional data is identified correctly?
Minor comments:
- The abbreviation for the Social Media Mining Toolkit should be introduced in the main text.
- It would be helpful if the authors could clarify whether the total number of annotated terms reported in Table 1 includes only drug-related terms or not?

---

> ### Author Response · Authors · 2020-09-26
> **Great points, there is definitely a need for further discussion of the differences between misspelling methods**
>
> We would like to thank the reviewer for the through and well described review. Here are our comments on your main concerns:
>
> As for your main concern: We are in complete agreement that the title of the paper is misleading and we will change it to better represent its contents. The current title represents that initial goals were more ambitious than the actual work done in this paper. As for the emphasis on the use of machine learning, we will change the text to play down its use and rather mention that our experiments contrast one machine learning method with other non-ML simpler methods. We understand the need to explain the observed differences between methods and we will add a paragraph and a few citations to address this.
>
> For the particular comments:
> 1) We mean letter, and this will be corrected. Digits are not traditionally used. We will add a clarification on how this is done and cite a relevant paper that analyzes this.
>
> 2) Table 2 shows the totals without any consideration for misspellings. We will clarify the table legend to be explicit about this.
>
> 3) Agreed.
>
> 4) We will gladly add commentary on this and provide a few examples as mentioned, for additional clarity and readability.
>
> 5) The misspellings bring in around ~70K additional tweets. We performed this step several times and performed a frequency analysis of the counts of the newly recovered terms, here we found some generated misspellings that actually represented another word and removed those for the next iteration.
>
> Minor comments:
> a) We will introduce the abbreviation, thanks for catching this.
> b) The dictionary we used only has RxNorm terms which includes only drug-related terms. However, there are a few ambiguous words that we have removed in our previous works.
>
> Thank you again for the detailed feedback.

---

### Official Review · AnonReviewer3 · 2020-09-22
**The paper is interesting, well presented, and concise with need for some corrections/clarifications.**

**Rating:** 7
**Confidence:** 4

**Review:**

The paper is about creating variations of COVID-19 drugs in Twitter using three different methods to capture posts with misspelled drug names.

Comments:

The Title and the Abstract gives the impression that paper is about finding potential COVID treatments in social media. However, the research is about the use of automatic techniques for dealing with misspelled drug names in Twitter.

The Number of English tweets used in this study and the duration of data collection needs to be mentioned. It is mentioned as 424 million, where only 67% is English. Its not stated how many of the tweets went through labelling process for COVID drugs.  If we only are targeting English tweets, then what is the point of labelling all 424 million tweets?

In the phrase “ For each term, each alphabet…” its more clear to use character/letter instead of alphabet.

Explanation of the third methodology seems to assume prior knowledge of the technology used.

Text tagging and annotation is used interchangeably, and I suggest using just tagging as annotation could imply manual effort in examining and labelling the tweets.

Zinc is stated as being irrelevant whereas it is talked about as enhancing the clinical efficacy of chloroquine.

Table 1 is unclear as what the “total annotated terms” represents? Is it the number of tweets?  And what is delta in the last row showing?

In Table 2 the total frequency is around 472,000 so where did the rest of 723,129 tweets with mentions of drug go? It also would be helpful to show the total number of misspelled drug names frequency.

In table 5, the calculation of percentage increase column needs clarification. The total of 222,428 is calculated as 15% of what number?

---

> ### Author Response · Authors · 2020-09-26
> **Great points and very useful feedback**
>
> We would like to thank the reviewer for the through and well described review. Here are our comments on your main concerns:
>
> 1) Paper's title: We completely agree with this, as other reviewers have also pointed this out. We will change it to better represent the contents of the paper.
>
> 2) Agreed, we started with the full set, and automatically tagged the full set using the top 10 languages, however, for this study we only kept and used the English tweets. We will clarify this statement.
>
> 3) Agreed, we will replace.
>
> 4) We shall add more discussion about this methodology to make the paper more self-standing.
>
> 5) Yes, we will standardize the use of text tagging and annotating. Thanks for pointing this out.
>
> 6) Indeed, we didn't consider Zinc initially, but as you mention it is part of a combination give. We will adjust our discussion for this.
>
> 7) Table one represents the total number of actual annotations made (on the English speaking tweets) the delta represents the number of extra terms found after running the error correction method on a partial set of tweets. We will certainly word this better for clarity.
>
> 8) The 'delta' from the total of table 2 is all other drugs and RxNorm terms that are found, which are less frequent. We can certainly calculate the frequencies.
>
> 9) This % comes from the difference of tagged tweets without any misspelling corrections and after applying the  misspelling analysis. We will clarify this language in the paper.
>
> Thank you again for the detailed feedback.